# Post-Keratoplasty Microbial Keratitis in the Era of Lamellar Transplants—A Comprehensive Review

**DOI:** 10.3390/jcm13082326

**Published:** 2024-04-17

**Authors:** Joanna Przybek-Skrzypecka, Katarzyna Samelska, Agata Joanna Ordon, Janusz Skrzypecki, Justyna Izdebska, Marta Kołątaj, Jacek P. Szaflik

**Affiliations:** 1Department of Ophthalmology, Medical University of Warsaw, Marszałkowska 24/26, 00-576 Warsaw, Poland; samelskakatarzyna@gmail.com (K.S.); justyna.izdebska@wum.edu.pl (J.I.); jacek.szaflik@wum.edu.pl (J.P.S.); 2SPKSO Ophthalmic University Hospital in Warsaw, 00-576 Warsaw, Poland; agata.j.ordon@gmail.com (A.J.O.); marta.kolataj@wp.pl (M.K.); 3Department of Binocular Vision Pathophysiology and Strabismus, Medical University of Lodz, 90-647 Lodz, Poland; 4Department of Experimental Physiology and Pathophysiology, Medical University of Warsaw, 00-576 Warsaw, Poland; janusz.skrzypecki@wum.edu.pl

**Keywords:** post-keratoplasty infectious keratitis, microbial keratitis, interface keratitis, corneal transplant, corneal ulcer, PK, DSAEK, DALK, DMEK, diagnosis

## Abstract

Microbial keratitis in a post-transplant cornea should be considered a distinct entity from microbial keratitis in a non-transplant cornea. Firstly, the use of immunosuppressive treatments and sutures in corneal transplants changes the etiology of keratitis. Secondly, corneal transplant has an impact on corneal biomechanics and structure, which facilitates the spread of infection. Finally, the emergence of lamellar transplants has introduced a new form of keratitis known as interface keratitis. Given these factors, there is a clear need to update our understanding of and management strategies for microbial keratitis following corneal transplantation, especially in the era of lamellar transplants. To address this, a comprehensive review is provided, covering the incidence, risk factors, causes, and timing of microbial keratitis, as well as both clinical and surgical management approaches for its treatment in cases of penetrating and lamellar corneal transplants.

## 1. Introduction

The growing number of contact-lens-wearers contributes to a higher incidence of microbial keratitis [1]. Infectious keratitis (IK) poses a significant threat for corneal transparency. An insufficient response to pharmacological treatment leads to therapeutic or optical corneal transplantation. The transplanted corneas are prone to further microbial invasions [2]. In a vicious circular mechanism, post-keratoplasty microbial keratitis (PKMK) poses risks for both vision and transplant function, with patients often requiring another keratoplasty. Furthermore, the number of available corneal tissues is limited and vastly lower than the current demand [3]. On top of this shortage of tissues, conservative treatments can be ineffective (due to the growing number of multidrug-resistant microorganisms) and people are often reluctant to donate their organs [4,5]. A global eye banking study showed that on average, 1 in 70 patients receives the transplant they need [3].

The annual number of corneal transplants and the paradigm of corneal transplantation have changed significantly in recent years [6,7]. Over a decade ago, the number of endothelial keratoplasties (DSAEKs and DMEKs, Descemet’s stripping automated endothelial keratoplasty and Descemet’s membrane endothelial keratoplasty, accordingly) outnumbered the number of penetrating keratoplasties performed globally [8]. The number of posterior lamellar keratoplasties is still increasing, altering the profile of typical PKMK [9,10]. Additionally, endothelial transplants have brought about interface keratitis (IK), with its unique etiology and management [11].

The natural history of microbial keratitis in patients with corneal transplants often leaves them with a low visual acuity [4,12]. PKMK often (up to 67% by *Dohse*) leads to graft failure [13]. More than half of grafts eventually lose transparency [14,15]. Overall, legal blindness afflicts around 70% of medically treated PKMK patients [12]. The question arises if we have the tools to stop this vicious circle.

This study aims to analyze the predisposing factors, etiology, rate of recurrence, and clinical treatment of, as well as surgical interventions for, microbial keratitis in patients with corneal transplants, divided into full-thickness corneal transplants and lamellar transplants. Additionally, interface keratitis was studied separately. Furthermore, a concise summary of available evidence is needed to council patients.

## 2. Methods

PubMed database search for articles regarding post-keratoplasty infectious keratitis was performed. Our study included all types of corneal transplants (penetrating, deep anterior lamellar keratoplasty (DALK), DSAEK, and DMEK) as well as peer-reviewed articles: original research, reviews, and case reports. Key words applied in the search comprised “post-keratoplasty infectious keratitis”, “post-keratoplasty microbial keratitis”, “corneal transplant infection”, “interface keratitis”, “microbial keratitis following corneal transplant”, and “microbial keratitis”. Articles in English published between January 1980 and November 2023 were evaluated. Data on demographics, risk factors, types of keratoplasties, time from transplantation to infection onset, microorganisms isolated (also divided into Gram-positive, Gram-negative, and fungi groups), rate of infected grafts in different corneal transplant types, final visual acuity (VA), rate of following infection surgical procedures (therapeutic grafts, evisceration, and enucleations), rate of endophthalmitis, and type of medical management were examined. Manuscripts that met our criteria were compiled in an Excel “16.0” (Microsoft, Redmond, WA, USA) spreadsheet and analyzed.

## 3. Results

### 3.1. Definition

Microbial keratitis is historically defined as a stromal inflammatory infiltrate of an infectious origin [16]. The vast majority of post-keratoplasty corneal infections start in the epithelium. Epithelial defects serve as a window for stromal invasion. However, this definition was extended when lamellar corneal transplants emerged, bringing about interface keratitis (IK) [17]. Notably, both the morphology and origin of PTMK differ between deep anterior lamellar keratoplasty (DALK) and endothelial keratoplasties (Descemet’s stripping automated endothelial keratoplasty (DSAEK) and Descemet’s membrane endothelial keratoplasty (DMEK)). Single stromal infiltrates still dominate in all types of transplants, but endothelial keratoplasties also present frequently with multifocal infiltrates located in the interface of donor and recipient corneas. A significant part of PKMK starts on the border of donor/recipient corneas. Chen et al. reported that 45% of microbial keratitis infections following penetrating keratoplasties are located at the donor–recipient junction [15].

### 3.2. Risk Factors

There are several risk factors for infectious keratitis in corneal transplantation. These can be categorized into four groups: host ocular status, the general health of the patient, surgery factors, and post-surgical management. A large body of evidence underlines the crucial impact of host status for dry eye, trichiasis, and persistent epithelial defects, which are key risk factors [8,18,19]. Moreover, patients with topical antiglaucoma agents, those who have had previous antiglaucoma surgeries, and those who have a history of Herpes infections should also be monitored [20,21,22]. Ulcer status must be assessed by looking at the following bad prognostic factors: fungal etiology, an ulcer size > 60 mm, limbal involvement, endothelial exudates, retro-iris exudates, corneal perforation, coexisting endophthalmitis, and a graft size ≥ 10 mm [23,24,25]. Furthermore, keratoplasty status prior to the infection needs to be mentioned as the infection in previously failed and decompensated graft has worse prognoses [26,27]. A transplant performed due to infectious keratitis also increases the risk of PKMK [28]. Finally, reduced corneal sensitivity contributes to a high risk of PKMK [29].

It is noteworthy that the general health of the patient might predispose to PKMK with diabetes mellitus, atopy (and other autoimmunological diseases), neoplastic disorders, and the advanced age of the recipient also plays a key role [6,30]. Additionally, alcohol-use disorder and low socio-economic status correlate with a higher ratio of PKMK [8,31].

Surgical factors concentrate vastly on a transplant type (penetrating keratoplasty brings higher risk thank lamellar keratoplasties) and the presence of sutures and their flaws (loose and broken sutures, unburied knots) [32,33]. Loose epithelium and large bandage contact lens also bring unfavorable outcomes. Secondary wound dehiscence is another risk factor for PKMK [12]. Last but not least, contaminated corneoscleral material is also an issue. Large-scale research reported the incidence of 0.25–5% of contamination despite the procedures that limit the potential transmission [34]. This problem affects posterior lamellar keratoplasties more often than penetrating and DALK transplants [35].

Post-surgical risk factors comprise a prolonged continued use of topical steroids, a prolonged extended use of antibiotics, prolonged CL wear, a topical glaucoma treatment, and an increased intraocular pressure [36,37,38].

Table 1 depicts the list of risk factors for PKMK organized in the five groups mentioned above.

### 3.3. Rate

Microbial keratitis affects 0.8–13% of all corneal grafts [39]. The incidence of PKMK varies among countries, with an indication of the transplantation, transplant type, and post-keratoplasty treatment protocol. Data on anterior lamellar keratoplasties with single reports or cohort studies with both penetrating and DALK patients are scarce. On the contrary, several large reports on the incidence of PKMK in posterior lamellar keratoplasties range between 0.01% and 0.92% [27,40,41]. Table 2 summarizes the incidence of PKMK in original papers published in the last 20 years.

It is well documented that developing countries pose a higher risk of PKMK due to agricultural risk factors which are common and a low socio-economic level [31]. Notably, a higher rate was also observed in failed grafts [27,42]. A vast majority of research proves a significant predominance of penetrating keratoplasties in PKMK. Posterior lamellar transplants pose a lower risk than PK and ALK of a subsequent microbial ulcer with the average rate of about 0.8% in one-year observation [27]. As an exception, an American study of 36 PKMK patients by Edelstein reported a higher rate of PKMK in endothelial grafts. However, Edelstein’s study considered donor-transmitted cases solely [43]. Interestingly, it provoked a discussion on the mandatory supplementation of antifungal agents (to the corneal tissue medium), but there is insufficient evidence for lowering the rate of fungal PTMK [40]. Also of note, posterior lamellar grafts pose a 1.5–3.0 times higher risk of fungal infection than PK and DALK, presumably associated with different tissue preparation conditions in the eye bank [43].

**Table 2 jcm-13-02326-t002:** Rate of post-keratoplasty microbial keratitis divided into penetrating and lamellar keratoplasties, abbreviations: PK: penetrating keratoplasty, DALK: deep anterior lamellar keratoplasty, DSAEK: Descemet stripping automated endothelial keratoplasty, DMEK: Descemet’s membrane endothelial keratoplasty, K: keratoprosthesis; n: number of eyes with post-keratoplasty microbial keratitis in the study, N: number of eyes with transplanted cornea in the study. **Studies comprising at least 10 episodes of PKMK in a posterior lamellar transplant are in bold**.

Study, 1st Author and Year	Country	n PKMK	N Grafts (+K)	PKMK Rate [%]	PK [%]	DSAEK [%]	DMEK [%]	DALK [%]
Sati 2022 [44]	India	31	789	3.9				
Dohse 2020 [13]	US	86	2098	4.1	5.9	1.3		
Zafar 2020 [27]	US	467	58,085	0.8		0.8		
Griffin 2020 [6]	UK	72	1508	4.8				
Okonkwo 2018 [22]	UK	59	759	5.4				
Quilendrino 2017 [9]	The Netherlands	2	500	0.4			0.4	
Sun 2017 [20]	Taiwan	67	871	7.7	7.7			
Chen 2017 [15]	Taiwan	42	648	6.5	6.5			
Edelstein 2016 [43]	US	66	354,930	0.02				
Constantinuo 2013 [26]	Australia	122	650	18.8	18.8			
Wagoner 2007 [45]	US	102	2103	4.9				
Tavakkoli 1994 [14]	US	36	885	4.9	4.9			

### 3.4. Etiology

The etiology of PKMK varies due to different types of corneal transplants and regions of the world [14,45,46,47]. The majority of research comes from the USA, western Europe, and India, so the results might be biased by an unequal representation of studies throughout the world. Furthermore, the etiology pattern has undergone some changes in the last three decades reflecting the shift in the corneal transplant type and treatment [44,48,49].

The available literature underlines bacteria as the most common origin of PKMK in PK and DALK, with a Staphylococcus preponderance [12,14,45,46]. Gram-positives outperform Gram-negatives in the majority of research, with some exceptions, e.g., Tavakkoli proved their equal incidence [14]. Of note, some studies, mainly of Asian origin, show a significant incidence of Pseudomonas aeruginosa infections (38% of all PKMK in the Taiwanese study by Chen, 29% in the Singaporean study by Ti) [15,47]. Also, Constantinuo et al. proved another Gram-negative bacteria—Moraxella—to be the most common etiological factor in failed grafts [26]. The difference from the mainstream trend was also reported in sutureless grafts, with a lower prevalence of Gram-positive cocci [32]. It is noteworthy that grafts after suture removal are frequently affected by Moraxella and Pseudomonas Gram-negative bacteria [32].

Another explanation for this etiological shift towards Gram-negatives might be related to the increased contact lens (CL) wear. Pseudomonas aeruginosa dominates in the prevalence of CL-related keratitis [1].

Fungal infections account for less than 20% of PKMK and occur more often in warm and humid environments. Candida preponderance is highly documented but Fusarium, Aspergillus, and Cryptococcus infections were also noted [15,44,50]. Protozoa infections following keratoplasty are relatively rare, but the high rate of recurrence must raise suspicion of Acanthamoeba infections [51].

PKMK in DSEAKs and DMEKs are mainly of bacterial and fungal origin, with a *Candida* spp. predominance [43]. Table 3 summarizes the literature on the etiology of post-keratoplasty infectious keratitis.

Finally, significant rate of negative scrapes’ results must also be underlined [24].

### 3.5. Interface Keratitis

Interface keratitis (IK) is a relatively rare complication of any lamellar corneal transplant, with dominant fungal etiology [17,52,53,54]. The largest study by Augustin shows a 0.15% rate of IK following DMEKs (n = 3950) [17]. A slightly higher prevalence of 0.92% was found by Nahum in the Italian cohort of 1088 DSAEK transplants (10/1088) [41]. According to a meta-analysis of Gao, 75% of infectious interstitial keratitis are of fungal origin with a *Candida albicans* preponderance [11]. There are scarce data on ALK presenting interface keratitis, as the presence of sutures predisposes to “typical” corneal ulcer- localized in corneal stroma [55]. However, a few case reports indicate the dominance of fungal etiology in IK following DALK [40,55]. On average, interstitial keratitis occurs early (typically within a few days after the transplant, mostly within 3 months), and in 80% of cases it roots from a donor-to-host transmission [17,56]. The potential risk of endophthalmitis and losing sight urge for an aggressive treatment [54]. Mostly, IK requires a surgical intervention, mainly re-transplantation. Alternatively, intrastromal injections of antifungals combined with interface infusion or interface drainage with antimicrobials might be sufficient to clear the infection [54,57,58].

**Table 3 jcm-13-02326-t003:** Etiology of post-transplant microbial keratitis reported in original research from 1990 to 2023 (of at least 30 cases), in chronological order.

Authors	Transplant Type	n	V	G+	G−	F	A	MIX	NG
Veugen, 2023 [59]	PK	829	52.7	32.7			14.6		
Sati, 2022 [44]	PK/DSAEK	31				16.2			
Dohse, 2020 [13]	PK/EK	75 PK	0	44	21.3	10.7	0	0	24
		11 EK		45.5	18.2	9.1	0	0	27.3
Griffin, 2020 [6]	PK/DALK/E	72	0	73	23	4	0	0	0
Ozalp, 2020 [60]	PK	36	41.7	38.9	16.7	2.8		11.1	
Okonkwo, 2018 [22]	PK/DALK	59	0	30.5	18.6	8.5	0	0	42.4
Sun, 2017 [20]	PK	67	0	58	22.4	19.7	0	11.5	0
Edelstein, 2016 [43]	PK/EK	66	4.5	3.2	4.5	51.8			36
Lin, 2016 * [61]	PK	50	0	50	42	20	0	12	0
Constantinou, 2013 [26]	PK	51	0	49		2	0	33.3	15.7
Wagoner, 2007 * [45]	PK	149	0	57	11.4	1.3	0	25.4	30.2
Vajpayee, 2002 [12]	PK	50	0	58	12	6	0	10	14
Tavakkoli, 1994 [14]	PK	36		47	47				
Bates, 1990 [48]	PK	30	0	63.3	20	13.3	3.3	3.0	0

Abbreviations: PK: penetrating keratoplasty, DALK: deep anterior lamellar keratoplasty, EK: endothelial keratoplasty, E: epikeratophakia; n: number of eyes with post-keratoplasty microbial keratitis in the study, V: ratio of viral etiology [%], G+: ratio of Gram-positive bacteria etiology [%], G−: ratio of Gram-negative bacteria etiology [%], F: ratio of fungal etiology [%], A: ratio of Acanthamoeba etiology [%], mix: ratio of mixed or polymicrobial infection [%], NG: no growth or scrapes not taken, empty block—unknown, * might exceed 100% because organisms were analyzed (mixed infection not separated).

### 3.6. Keratitis in Repeated Grafts

Scarce data are available concerning the second and subsequent infections in corneal grafts. Worse clinical outcomes are associated with each subsequent corneal transplant. There are several well-established risk factors for recurrent infections in therapeutic grafts: fungal origin, big ulcer size (diameter of at least 60 mm), graft size (diameter of at least 8.5 mm), endophthalmitis, limbal involvement, perforated cornea, endothelial or retro-iris exudates [23]. Still, the most common origin of repeated microbial infections is bacteria (>50%), followed by viruses and fungi [60]. The recurrence rate varies from 2–3% to 41% in different analyses [2,23,59,62]. A subsequent ulcer is detected early, typically within the first two to three months, after the transplantation in repeated grafts (mean 16 days according to Chatterjee et al.) [23,47]. The robust analysis by Veugen has ascertained that fungal keratitis is associated with highest probability of recurrence in the therapeutic graft (15%), as well as the shortest interval between the grafts [59,63]. Also, Acanthamoeba keratitis often recur (8–20%), followed by viral and bacterial keratitis [51,59]. Acanthamoeba tends to store cysts in the recipient cornea, frequently close to the limbus, leaving a high rate of potential flare-up. (Australian Registry 2015) Previous glicocorticosteroid treatment and hypopyon are well-established risk factors of Acanthamoeba recurrence in the graft [64]. Several reports proved the need for up to six subsequent keratoplasties for Acanthamoeba re-infections [65]. Of note, both penetrating and deep anterior lamellar keratoplasty are applied in therapeutic transplants for Acanthamoeba infection, while fungal re-infection always requires a full-thickness corneal transplant [66].

### 3.7. Time

The time between transplantation and the occurrence of PKMK plays a crucial role in appropriate management and the final visual outcome. The range from 1 day to several years after corneal transplant has been described, with an average of 12–30 months [6,67]. Of note, PMKM in previous therapeutic grafts occurs significantly earlier [23]. Table 4 summarizes the available literature on the time of PKMK onset after corneal transplant derived from the last 18 years of original research. Notably, fungal etiology brings about bacterial and viral infections [60]. Interestingly, contaminated donor tissue tends to manifest corneal ulcer early (within the first 2 months) [56]. Also, failed grafts are associated with prompt PKMK [26].

#### 3.7.1. Intraoperative Infection and Contaminated Donor Tissue

Despite the procedures limiting the potential transmission of the donor-to-host infection, microbial keratitis might occur due to an infection of the donor tissue. According to the literature, posterior lamellar keratoplasties bring about a higher risk [35]. Fungal keratitis (*Candida albicans*) seems to be the most prevalent pathogen [35].

#### 3.7.2. Immunosuppressive Treatment

The gold standard of post-keratoplasty treatment comprises topical glicocorticosteroids (GCs) that might contribute to an increased risk of refractory infection in the transplant. There is no one universe protocol for the type and frequency of GCs applied. They encompass fluorometolone, loteprednol, and prednisolone acetate, the latter being the most commonly applied. [13] However, they are stopped or tapered after PKMK in the majority of cases. Wagoner suggests their immediate withdrawal when PKMK occurs and gradual introduction after 48–72 h of antimicrobial treatment according to the etiology (48 h in Gram-positive and 72 h in Gram-negative, mixed bacterial, and culture-negative clinically presumed bacterial PKMK) [45]. Special concern is given to fungal infections which require 1–2 weeks intervals between therapeutic PK and the safe introduction of topical glicocorticosteroids [69,70].

On the other hand, omitting anti-inflammatory drops poses a threat of early graft rejection/decompensation or prolonged anterior segment inflammation which also increases the risk of PMKM [69].

There is no simple solution to the gain and loss of glicocorticosteroidal anti-inflammatory treatment, but several alternatives have been investigated. First of all, other immunosuppressive specimens have been widely studied, starting with cyclosporin 0.1% which has proven efficacy as a post-keratoplasty anti-inflammatory treatment [71]. Another one, tacrolimus, the second calcineurin inhibitor with a lower potential for impeding corneal epithelium, has been explored [72].

Special concern is given to a herpetic infection. Possible difference among the studies concerning post-keratoplasty herpetic keratitis may be a result of discrepancies in the use of antiherpetic drugs. Currently, in many clinics, there is no standardized protocol for oral antiherpetic drugs after keratoplasty performed in the eyes after a preliminary herpetic infection. A systematic review summarizing results of acyclovir use has implied that in people having keratoplasty due to HSV infection, oral acyclovir use may lower the risk of herpetic keratitis and graft failure [73]. Both of these implications rely on low-certainty evidence. The studies analyzed suggest the dose of oral acyclovir should vary between 200 and 800 mg administered 1–3 times daily. The suggested time of the treatment also varies.

### 3.8. Treatment

Irrespective of anti-inflammatory treatment, infectious keratitis in corneal transplants requires antimicrobial agents. PKMK’s treatment starts with topical anti-infectious agents. The gold standard antibacterial regimen consists of either fluoroquinolones in monotherapy or fortified cephalosporines combined with fortified aminoglycosides [74]. There is no difference in the outcome between those two strategies. However, ofloxacin monotherapy poses a lower risk for epithelial changes, chemical conjunctivitis included [75]. Of note, topical vancomycin replaces topical cephalosporines due to their unavailability in some parts of the world. Antifungal management requires topical voriconazole, amphotericin B, or natamycin depending on the fungal genre and it might be combined with parenteral voriconazole administration.

However, several limitations pose a challenge to effective treatment. Firstly, antibiotic-resistant organisms have increased in the last few years [5]. Global reports have proved the rising tide of resistance to methicillin Gram- positive bacteria, e.g., Staphylococcus aureus including multidrug resistance comprising fluoroquinolones, aminoglycosides, and macrolides [5].

Additionally, access to compounded medications is limited, especially in rural areas [76]. Secondly, the improvement of active substances as well as preservatives’ side effects is also needed. Thus, the novel formulations of available active substances are also studied. These formulations encompass a range of drug delivered systems such as inserts, nanoparticles, liposomes, niosomes, cubosomes, microemulsions, in situ gels, contact lenses, nanostructured lipid carriers, carbon quantum dots, and microneedles. Ex vivo and in vivo studies have demonstrated the prolonged residence time of active substances in the cornea and the increased ocular bioavailability. These formulations have also shown successful treatment of keratitis in animal models, primarily focusing on fluoroquinolones for bacterial keratitis and limited research on antifungal, viral, and Acanthamoeba keratitis [77,78]. Of note, contact lenses offer a unique platform for delivering drugs directly to the eye, providing localized and sustained release of medications, their excellent bioavailability, and an increased ocular drug residence time [79]. Furthermore, photo-activated chromophores for infective keratitis cross-linking (PACK-CXL) have been studied in halting microbial keratitis progression, giving promising results [80]. Finally, a novel treatment called Rose Bengal-mediated photodynamic therapy with green light optical irradiation (RB-PDAT) needs to be mentioned. Primarily established to cease progressive keratitis and minimize the therapeutic keratoplasty rate, it might be also implemented in the PKMK treatment armamentarium [81].

The increasing demand for and limited supplies of donor corneas from the eye banks have contributed to long waiting lists for corneal transplantation in most developing countries. The shortage of human corneas has led to some major developments in the field of corneal tissue engineering, particularly in the use of silk films as substrates to grow corneal cells for ocular surface reconstruction at any of the cornea’s layers [82].

### 3.9. Surgical Interventions

Despite an optimal pharmacological therapy, surgical management of PKMK is crucial in saving the eyeball or preventing the infection from spreading beyond the orbit [83,84,85]. Several risk factors of surgical interventions were previously established: perforations, old age, non-healing epithelial defects, and previous ocular surgery [86,87]. The mainstay of surgical interventions in PKMK includes therapeutic and optical corneal transplants [62]. In general, therapeutic keratoplasty has worse outcomes (shorter time, higher rate of graft failure) than optical and tectonic transplants. The main indications of the therapeutic method include a corneal perforation of at least 3 mm size, and progressive MK despite maximum pharmacological treatment and infection associated with severe, progressive thinning [88,89]. However, interface keratitis might also require a therapeutic graft, especially of a fungal origin [41]. Despite penetrating keratoplasty being the gold standard of therapeutic keratoplasty, we need to also report DALK as being an alternative [90]. On the contrary, a regraft of posterior lamellar transplants should be avoided due to the high recurrence rate of infection [41,91]. Another option for whole cornea ulcer or vast perforation is a corneoscleral transplant [92]. However, the global shortage of tissue imposes consideration of alternative methods of non-pharmacological treatment. Thus, the surgical armamentarium for PKMK comprises amniotic membrane transplantation, glue applications, autologous tissue transplantations (mucous, conjunctiva, tendon patch, skin transplant), or even bandage contact lens [85]. A corneal lenticule obtained through refractive surgery (SMILE—small incision lenticule extraction—procedure), which is relatively new, might also be an alternative when available [93]. Another procedure of tarsorraphy is applied in patients with a persistently compromised epithelium and thus ineffective corneal healing, e.g., Stephen-Johnson syndrome [83,94]. Single reports on compression sutures on dehiscent wounds also show effectiveness in preventing the infection from entering the eye [95].

Special interest was given to the rate of irreversible surgical treatment of evisceration or enucleation. Endophthalmitis is the main risk factor, followed by elderly age, glaucoma, prior ocular surgery, ocular surface disease, topical steroids, and systemic diseases [96]. Fungal etiology of endophthalmitis conceals the most prevalent one [84]. Coagulase-negative streptococcus and Pseudomonas aeruginosa among Gram-negative bacteria prevail in most regions of the world [87]. Its rate also differs between the world’s regions with the trend of higher preponderance in developing countries, e.g., India [23,96]. Table 5 summarizes the available data on regraft and eyeball removal rates due to infectious keratitis in corneal transplants.

### 3.10. Outcomes

PKMK increases the risk of transplant failure 2–3 times more in PK than after endothelial keratoplasty [13]. The majority of them are successfully managed medically. However, 3–41% require therapeutic keratoplasty and 2–21% result in eyeball removal. Finally, 60–70% of full-thickness grafts lose transparency following PKMK [15,18,20,62]. Clear grafts’ rate is higher after endothelial keratoplasty infection than penetrating keratoplasty (67% vs. 28% according to Dohse) [13]. Glaucoma increases the proportion of failed grafts [20]. Considering etiology, Acanthamoeba and fungi show the worst survival outcome [13,59]. Visual acuity differs significantly between world regions, the type of the transplant, and its pre-infection status. Functional visual acuity of at least 100/200 is reached in around 8% of cases [13,45]. On the contrary, counting fingers or a lower VA account for 40–50% of PKMK patients [13,15,45].

## 4. Conclusions

Post-keratoplasty microbial keratitis poses a significant threat to transplant clarity and subsequently visual acuity. The era of lamellar transplants has reduced the overall risk of PKMK. The lowest rate of PKMK is noted in DMEKs (0.4%), followed by DSEAKs cohorts (0.8–1.2%), whereas the risk in penetrating keratoplasty and DALKS is calculated to be on average 4–6% with higher numbers in some developing countries. On average, corneal ulcers occur between 12 and 30 months after the transplantation, except donor-transmitted infections and interface keratitis presenting within the first 2 weeks from the surgery. Bacterial etiology dominates in penetrating and anterior lamellar keratoplasties, whereas the fungal origin is found most often in interface keratitis following posterior lamellar transplants. The well-established risk factors of PKMK encompass the older age of the recipient, fungal etiology, previously failed graft, previous infectious keratitis, and coexisting glaucoma. A surgical intervention is required in 3–41% of cases. This includes therapeutic full-thickness corneal transplant for medically uncontrolled infections, interface keratitis included, as well as eviscerations for PKMK progressing to poorly managed endophthalmitis. The vast majority of patients end up with a visual acuity of less than 20/40 (around 90–92%), with a final legal blindness (<20/200) rate of 70–80%. The provided dataset enables proper patient counseling on the potential course of the disease.

## Figures and Tables

**Table 1 jcm-13-02326-t001:** Risk factors for infectious keratitis in corneal transplants.

Group of Risk Factors	Risk Factor
Host ocular status	dry eye trichiasis persistent epithelial defect topical and surgical glaucoma treatment history of Herpes infections ulcer status (fungal etiology, ulcer size > 60 mm, limbal involvement, endothelial exudates, retro-iris exudates, corneal perforation, coexisting endophthalmitis, graft size ≥ 10 mm) previously failed and decompensated graft transplant performed due to infectious keratitis reduced corneal sensitivity
General health of the patient	diabetes mellitus neoplastic disorders atopy and other autoimmunological diseases advanced age of recipient
Transplant status	transplant type (a higher risk in penetrating keratoplasty) loose and broken sutures, unburied knots loose epithelium large bandage contact lens wound dehiscence contaminated corneo-scleral material
Post-surgical management	prolonged continued use of topical steroids prolonged extended use of antibiotics prolonged contact lens wear topical glaucoma treatment increased intraocular pressure

**Table 4 jcm-13-02326-t004:** The time of the onset of post keratoplasty infectious keratitis based on the eligible studies from 2006 to 2022, PK: penetrating keratoplasty, EK: endothelial keratoplasty, n: number of eyes with post-keratoplasty microbial keratitis in the study, N: number of eyes with transplanted cornea in the study.

Authors, Year	Country	n (N)	Time from the Transplantation
Dave, 2022 [62]	India	78	12 months
Chatterjee, 2020 * [23]	India	50 (229)	14 days
Dohse, 2020 [13]	US	86 (2098)	29 months (PK) 30 months (EK)
Griffin, 2020 [6]	UK	72 (1508)	25 months
Moon, 2020 [2]	Korea	19	17 months
Okonkwo, 2018 [22]	UK	59 (759)	49.5 months
Chen, 2017 [15]	Taiwan	42 (648)	12 months
Sun, 2017 [20]	Taiwan	67/871	1 year (72% within 1 year)
Constantinou, 2013 [26]	Australia	122 (650)	69 months
Wright, 2006 [68]	US	44	26 months

***** A study of PKMK occurring only in therapeutic transplants due to microbial keratitis.

**Table 5 jcm-13-02326-t005:** Surgical intervention in microbial keratitis in corneal transplants, divided by world region, data from the years 2002 to 2022. n: number of eyes with post-keratoplasty microbial keratitis in the study or summarized studies from one country; *: a study of PKMK occurring only in therapeutic transplants due to microbial keratitis.

Study	Country	n	Regraft Rate [%]	Evisceration/Enucleation Rate [%]
Dave 2022 [62] Sati 2022 [44] Raj 2018 [24] Valpayee 2002 [49]	India	216	9–41	2–10
Chatterjee 2020 * [23]	India (therapeutic tranplants)	63	3	21
Dohse 2020 [13] Wright 2006 [68]	US	130	29–34	0–6
Griffin 2020 [6]	UK	72	24	4
Moon 2020 [2]	Korea	19	32	11
Sun 2017 [20]	Taiwan	52	31	10
Constantinou 2013 [26]	Australia	51	20	2

## Data Availability

Not applicable.

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
