# Peer review of "Post-Keratoplasty Microbial Keratitis in the Era of Lamellar Transplants—A Comprehensive Review"

_jcm, 2024, doi:10.3390/jcm13082326_

Round 1

Reviewer 1 Report

Comments and Suggestions for Authors

In this manuscript, Joanna Przybek-Skrzypecka et al present a comprehensive review on post-keratoplasty microbial keratitis. The authors exhibit a deep understanding of the subject matter, evident from their adept description of the background in the introduction section, where they advocate for the significance of this research endeavor. Throughout the manuscript, the authors provide well-rounded information on different dimensions of microbial keratitis, with an emphasis on post-keratoplasty microbial keratitis. By dissecting various aspects of this condition, from its etiology to management strategies, the authors offer valuable insights that contribute significantly to the existing body of knowledge in the field.

Reviewer 2 Report

Comments and Suggestions for Authors

This manuscript provides a comprehensive overview of post-keratoplasty microbial keratitis in lamellar transplants. The manuscript could be improved by addressing the following issues:

1.     Introduction: It could benefit from a brief introduction to the different types of lamellar transplants.

2.     Page 4, it would enhance the clarity by including a table to explain the risk factors for infectious keratitis in corneal transplant.

3.     Page 6, line 166, reference [42] was cited for twice.

4.     Table 1, it could be beneficial to consolidate and summarize the data pertaining to the same county across various studies.

5.     Table 2, the authors could consider synthesizing and summarizing the data related to the same surgical procedure from various studies.

6.     Table 4, same as table 1, the authors could consider consolidating and summarizing the data about the same county across various papers.
